# Admixing Fir to European Beech Forests Improves the Soil Greenhouse Gas Balance

**Stephanie Rehschuh [1], Martin Fuchs [1,2], Javier Tejedor [1], Anja Schäfler-Schmid [1], Ruth-Kristina Magh [2] , Tim Burzlaff [2], Heinz Rennenberg [2] and Michael Dannenmann [1,\*]**

[1] Institute of Meteorology and Climate Research–Atmospheric Environmental Research (IMK-IFU), Karlsruhe Institute of Technology (KIT), Kreuzeckbahnstr. 19, 82467 Garmisch-Partenkirchen, Germany; stephanie.rehschuh@kit.edu (S.R.); unimartinfuchs@gmail.com (M.F.); jtejebis@hotmail.com (J.T.); anja.schaefler-schmid@kit.edu (A.S.-S.)

[2] Institute of Forest Sciences, University of Freiburg, Georges-Koehler-Allee 53/54, 79110 Freiburg, Germany; ruth.magh@ctp.uni-freiburg.de (R.-K.M.); tim.burzlaff@ctp.uni-freiburg.de (T.B.); heinz.rennenberg@ctp.uni-freiburg.de (H.R.)

\* Correspondence: michael.dannenmann@kit.edu; Tel.: +49-8821-183-127

**Abstract:** Research highlights: The admixture of fir to pure European beech hardly affected soil-atmosphere $CH_4$ and $N_2O$ fluxes but increased soil organic carbon (SOC) stocks at a site in the Black Forest, Southern Germany. Background and objectives: Admixing deep-rooting silver fir has been proposed as a measure to increase the resilience of beech forests towards intensified drying-wetting cycles. Hence, the goal of this study was to quantify the effect of fir admixture to beech forests on the soil-atmosphere-exchange of greenhouse gases (GHGs: $CO_2$, $CH_4$ and $N_2O$) and the SOC stocks by comparing pure beech (BB) and mixed beech-fir (BF) stands in the Black Forest, Germany. Materials and methods: To account for the impact of drying-wetting events, we simulated prolonged summer drought periods by rainout shelters, followed by irrigation. Results: The admixture of fir to pure beech stands reduced soil respiration, especially during autumn and winter. This resulted in increased SOC stocks down to a 0.9 m depth by 9 t C ha$^{-1}$ at BF. The mixed stand showed an insignificantly decreased sink strength for $CH_4$ ($-4.0$ under BB and $-3.6$ kg C ha$^{-1}$ year$^{-1}$ under BF). With maximal emissions of 25 µg N m$^{-2}$ h$^{-1}$, $N_2O$ fluxes were very low and remained unchanged by the fir admixture. The total soil GHG balance of forest conversion from BB to BF was strongly dominated by changes in SOC stocks. Extended summer droughts significantly decreased the soil respiration in both BB and BF stands and increased the net $CH_4$ uptake. Conclusions: Overall, this study highlights the positive effects of fir admixture to beech stands on SOC stocks and the total soil GHG balance. In view of the positive impact of increased SOC stocks on key soil functions such as water and nutrient retention, admixing fir to beech stands appears to be a suitable measure to mitigate climate change stresses on European beech stands.

**Keywords:** climate change; drying-wetting cycles; gas fluxes; mixed forest; carbon storage

## 1. Introduction

A global rise in temperature and an increasing frequency, duration and intensity of weather extremes are observed and will gain further importance within the next century [1]. A redistribution of summer precipitation is likely to occur in northern midlatitudes with prolonged summer droughts followed by heavy rainfall events [1]. Especially in southwestern Germany, including the Black Forest, such weather extremes are an increasing challenge [2,3]. These "drying-wetting cycles" will affect soil moisture and temperature regimes with potential impacts on the microbial nutrient turnover

and nutrient supply to trees, the soil-atmosphere exchange of greenhouse gases (GHGs) and the soil organic carbon (SOC) stocks [4–6]. This could be critical for forest stands because of their relatively small adaption capacity to irregularly occurring disturbances [7]. European beech (*Fagus sylvatica* L.) is one of the dominant species in Central Europe, being particularly sensitive to drought [8–10]. Due to its low potential to adapt, the geographic distribution of European beech will most likely decline and/or rather shift to northern latitudes [11–13]. In order to mitigate climate impacts on forests, substituting monocultures with mixed forests has been suggested to improve biodiversity and recovery rates [14,15]. For drought-sensitive beech, admixing deep-rooting trees such as oak or silver-fir have been proposed [16,17]. Silver fir is as an ecologically valuable and indigenous tree species in many European mountain forests [18]. It is expected to benefit beech by hydraulic lift and hydraulic redistribution of water, i.e., the rehydration of dry topsoil due to the passive movement of water originating from deeper soil layers from the roots to the dry topsoil, influenced by a gradient in water potential [19].

However, differences in the leaf litter quality and quantity, the canopy throughfall, the root system and the root activity can have profound influences on soil physical properties (e.g., moisture and temperature), soil chemistry, carbon (C) and nitrogen (N) cycling in the soil and the soil-atmosphere-exchange of GHGs [20–22]. An altered source or sink strength of soils for GHGs under different tree species was often found to be related to the input of litter and its characteristics [23,24], the permeability and morphology of organic horizons [20,25], the soil microclimate [24], the different soil acidification and concentration of inhibitory compounds [25], the nutrient availability for vegetation [26], as well as root-related processes such as water and nutrient uptake, root respiration or rhizodeposition [21,24].

Studies analyzing tree species' effects on soil respiration have revealed controversial outcomes: Some do not support the hypothesis that broadleaf stands emit more carbon dioxide ($CO_2$) from the soil [27–30]; others detected significantly higher soil respiration rates for deciduous stands compared to coniferous forests [24,31] with intermediate values under mixtures [31]. Several studies showed that soils under hardwood tree species consumed higher rates of atmospheric methane ($CH_4$) than coniferous forest soils as reviewed by Oertel et al. [32], thereby considering 18 studies. Borken and Beese [20] investigated soil-atmosphere $CH_4$ fluxes in mixed spruce and beech stands and detected a decreasing net $CH_4$ uptake with an increasing spruce presence in beech stands, whereas such an effect on nitrous oxide ($N_2O$) fluxes was not found. However, at the Höglwald site in Southern Germany, approx. 4 times higher mean annual $N_2O$ emissions from pure beech stands compared to adjacent pure spruce stands were measured [33]. Such studies are rarely available for mixed stands, and no study assessed the effect of fir admixture to beech forests on the GHG exchange at the soil-atmosphere interface.

The effects of drying-wetting cycles on soil-atmosphere GHG fluxes have been in the focus of numerous studies [34–37]. Drought generally reduced trace gas fluxes of $CO_2$ and $N_2O$ [35–37]. With regards to $CH_4$, a reduced soil water content will promote net $CH_4$ uptake due to the improved diffusion properties of dry soil (e.g., Reference [38]). However, $CH_4$ oxidizers can also suffer from drought stress under extreme drought conditions, which can lead to reduced net $CH_4$ uptake [36]. Also, rewetting experiments revealed ambiguous results: While soil respiration pulses exceeding the fluxes of constantly moist soil were observed [26], others [36,39] found no recovery of soil respiration after wetting of dry soil. Fluxes of $N_2O$ increased drastically after rewetting in some cases [37,40], which was contradictory with other studies [36] where GHG fluxes remained significantly reduced after rewetting of dry soil. Contrasting results show that the extent of drying-wetting pulses is not well-understood and largely diverse across different forest stands. The corresponding knowledge for mixed beech-fir stands in comparison to pure beech stands is missing.

Since mixed beech-fir forests are a promising silvicultural management measure in a changing climate, especially in sensitive regions such as the Black Forest in southwest Germany [41], in this study, we examined the influence of the admixture of fir to beech stands on soil-atmosphere GHG exchange ($CO_2$, $CH_4$ and $N_2O$) and soil carbon sequestration, thereby taking drying-wetting cycles

into account. We hypothesized that fir admixture would decrease the soil respiration, the net $CH_4$ uptake and the $N_2O$ emissions but would increase SOC stocks.

## 2. Materials and Methods

### 2.1. Study Site and Experimental Design

The measurements were conducted on a northwest-exposed hillside slope in the foothills of the southern Black Forest in the municipal of Freiamt close to Emmendingen, Germany (48°14´N, 7°90´E) at an elevation of 400–460 m a.s.l. (Table A1). The mean annual air temperature at the site is 9.6 °C on average (1971–2003). The annual rainfall is approximately 1020 mm (1971–2003) (station Freiamt-Ottoschwanden, Freiamt, Germany) with much precipitation occurring from May to July (37% of the total). The soil with a sandy loam texture was derived from red sandstone of the Buntsandstein formation and was classified as Dystric Cambisol according to the World Reference Base for Soil Resources [42]. The site has 40–60 years old pure beech and beech-fir mixed stands. Three pairs of adjacent pure beech (BB) and mixed beech-fir stands (BF) were selected for the experiments at different slope positions (middle slope, top slope and hilltop in flat terrain; Table A1). Rainout shelters of 200 m² (BB) and twice 100 m² (BF) were installed from July 19th until October 4th in 2016 and May 11th until August 18th in 2017 to fully exclude natural precipitation in order to induce summer drought. This resulted in four different treatments: two with a rainout shelter during the summer (BB drought/BF drought) and controls under ambient precipitation (BB control/BF control). After 2.5 months of rainfall exclusion in 2016, we applied 100 liters of water per m² using sprinklers. In 2017, after three months of drought, the soil was rewetted with $40\,l\,m^{-2}$ and additionally with $60\,l\,m^{-2}$ one week later. Irrigation lasted over 4–5 h. While the soil-atmosphere GHG flux measurements were restricted to these precipitation manipulation plots at the middle slope position, SOC stock quantification was conducted at a larger area at three slope positions to improve the spatial coverage (Table A1).

### 2.2. Meteorological Data

The daily precipitation was measured at a nearby climate station of the Energie Baden-Württemberg AG (EnBW Freiamt, 430 m) at a distance of 0.85 km to the plots. The data of air temperature (2 m height, measured in the stand), volumetric soil moisture and soil temperature were recorded every second hour by decagon EM50 loggers (Decagon Devices, Inc., Pullman, Washington, DC, USA). The missing values of air temperature before April 2016 were augmented by the data of the EnBW climate station Freiamt. For each treatment, the soil moisture was recorded by 4–5 TDR (time domain reflectometry) sensors (Type GS1 and 5TM, Decagon Devices, Inc., Pullman, WA, USA) and soil temperature sensors (Type 5TM), both installed at a 15 cm depth. The sensors were calibrated by the gravimetric measurements of soil samples. The water-filled pore space (WFPS) was calculated from Equation (1), where VWC = volumetric water content, BD = soil bulk density and DS = density of particles (assumed to be $2.65\,g\,cm^{-3}$ for minerals and $1.3\,g\,cm^{-3}$ for humus in the Ah and Bv1 horizon).

$$WFPS = \frac{VWC}{1 - \frac{BD}{DS}} \qquad (1)$$

### 2.3. Soil Organic Carbon Stocks

The soil sampling was conducted in March 2017 at replicated plots at different landscape positions (see above and Table A1). The soil pits were sampled to a depth of approximately 90 cm, which corresponded to the beginning of bedrock. Three replicated samples were taken from each horizon in each pit. Furthermore, 10 additional samples were taken randomly around the pits for the two upper horizons Ah (0–7.5 cm) and Bv1 (7.5–15 cm). Litter was collected quantitatively at ten randomly selected spots within the different treatments by using a 20 × 20 cm frame. The soil was dried at 60 °C until

constant weight, sieved (2 mm mesh) and ground using a ball mill (MM200, Retsch, Haan, Germany). The litter was equally dried, and the total dry weight determined before homogenization and grinding. The samples were sent to a commercial laboratory (Dr. Janssen, Gillersheim, Germany) for the determination of total carbon (TC) and carbonate using the VDLUFA Method A 5.3.1 for carbonate and DIN ISO 10,694 for TC. Since no carbonate was found, TC equaled SOC. For the quantification of SOC stocks, the soil bulk density was measured in three replicates per soil pit and the horizon by volumetric sampling, either with soil cores (volume of 100 $cm^3$) in the upper horizons or by quantitative collecting soil of 20 cm × 20 cm × depth (cm) of the specific horizon in the lower soil layers and subsequent quantification of soil weight after drying at 105 °C, thereby considering stone content. A total of nine samples were analyzed for texture using wet sieving and sedimentation (DIN ISO 11277) by Dr. Janssen´s laboratory (Gillersheim, Germany). For the estimation of changes in the carbon sequestration and soil GHG balance due to fir admixture, the difference of SOC stocks between BB and BF was used to calculate the annual change of SOC stocks as global warming potentials (GWPs), thereby considering the stand age of 50 years (Equation (2)):

$$CO_{2GWP} \left( kg \; CO_2 - eq \; ha^{-1} yr^{-1} \right) = - \frac{SOC \; stock \; BF - BB \; \left( kg \; C \; ha^{-1} \right) \; \times \; 44}{12 \; \times \; 50} \qquad (2)$$

where 12 is the molecular weight of C in $CO_2$ and 44 is the molecular weight of $CO_2$.

### 2.4. Soil-Atmosphere Exchange of Greenhouse Gases

The manual static chamber method was used for in situ measurements of soil-atmosphere GHG exchange ($CO_2$, $CH_4$ and $N_2O$) at the soil-atmosphere interface [43]. Sampling was carried out manually approximately weekly between February 2016 and November 2017. In each of the four treatments described above, five to six chambers were installed. The chambers were made out of polypropylene with dimensions of 37.0 × 26.5 × 11.5 cm. The chambers were equipped with septa and allowed for pressure equilibration via an 1/8-inch PTFE tube. To avoid a gradient in the gas concentration during sampling, the chambers were equipped with a fan (axial fan, Type: 412, EBM Papst, Mulfingen, Germany) connected to a 12-volt battery. A detailed description of the chambers can be found in Reference [44]. For the gas sampling, a 60 ml syringe (Omnifix ®, B. Braun, Melsungen, Germany) with Luer-Lock-valve (VWR International, Darmstadt, Germany) and injection cannula (Omnifix ®, B. Braun, Melsungen, Germany) was used. Four headspace samples of 60 ml each were manually sampled at 0, 10, 20 and 30 min after gas tight closure of the chambers. Sample air was then injected in 10 ml vials with a sealed lid (SRI Instruments, Bad Honnef, Germany), with the first 45 ml used for flushing of the vial and the remaining 15 ml being injected in the vial with overpressure. Concentrations of $CO_2$, $CH_4$ and $N_2O$ of sample air in the vials were determined by gas chromatography (GC) at the Institute of Meteorology and Climate Research, Atmospheric Environmental Research in Garmisch-Partenkirchen. The GC (8610 C, SRI Instruments, Torrence, CA, USA) was equipped with a flame ionization detector/methanizer (FID) for $CH_4$ and $CO_2$ detection and an electron capture detector (ECD) for analyzing $N_2O$. The system was calibrated using standard gas samples ($N_2O$: 406 ppb; $CH_4$: 4110 ppb; $CO_2$: 407.9 ppm, Air Liquide, Düsseldorf, Germany). The flux rates were calculated for each chamber from linear changes of gas concentrations over time (*n* = 4 sampling points in 30 minutes). The fluxes were corrected for temperature, and quality checks were applied for all gas measurements, while discarding all $CO_2$ and $CH_4$ fluxes with a value of $R^2$ smaller than 0.72. For $N_2O$, all fluxes exceeding +5 or −5 µg $N_2O$–N $m^{-2}$ $ha^{-1}$ had to match a quality criterion of $R^2 > 0.72$. To gain cumulative fluxes over the measurement period, the measured fluxes were projected to 24 h. Daily fluxes between the measurements were calculated from linear

interpolation. To obtain the non-$CO_2$ GHG balance, the GWPs for $N_2O$ and $CH_4$ were calculated from cumulative annual fluxes using Equations (3) and (4):

$$N_2O_{GWP}\left(kg\ CO_2 - eq\ ha^{-1}yr^{-1}\right) = \frac{N_2O\left(kg\ N_2O - N\ ha^{-1}yr^{-1}\right)}{28} \times 44 \times 265 \qquad (3)$$

where 28 is the molecular weight of N in $N_2O$ and 44 is the molecular weight of $N_2O$. The 100-year GWP of 1 kg $N_2O$ corresponds to 265 kg $CO_2$ [45].

$$CH_{4GWP}\left(kg\ CO_2 - eq\ ha^{-1}yr^{-1}\right) = \frac{CH_4\left(kg\ CH_4 - C\ ha^{-1}yr^{-1}\right)}{12} \times 16 \times 28 \qquad (4)$$

where 12 is the molecular weight of C in $CH_4$, 16 is the molecular weight of $CH_4$ and 28 is the $CO_2$-equivalient of 1 kg $CH_4$ based on 100 years [45].

*2.5. Data Presentation and Statistical Analysis*

All statistical analyses and data presentations were carried out with *R* 3.5.3 [46]. The Shapiro–Wilk test was applied to test for normal distribution, and the Levene test was used to test homogeneity of variances at $p < 0.05$. For data matching these criteria, the *t*-test was applied to identify significant differences between two groups. One-way analysis of variance (ANOVA) followed by the Tukey post hoc test was used to compare the mean values among various groups. Whenever the normal distribution of the data and the homogeneity of variances were not confirmed, the Wilcoxon–Mann–Whitney test (two groups for comparison) and the Kruskal–Wallis test followed by the Bonferroni post hoc analysis (several groups) were performed. For all statistical tests, $p < 0.05$ was used as the threshold for significant differences. Regression models (linear and 2nd order polynomial) were used to test for a correlation between soil-atmosphere GHG fluxes ($CO_2$, $CH_4$ and $N_2O$) and soil temperature and soil moisture in order to evaluate to which extent these parameters could represent predictive potentials for GHG fluxes. Further, multiple polynomial regressions considering both indicators were applied. For testing the influence of the interaction of the forest type (BB and BF) and the drought/control treatment on GHGs (simulated drought period fluxes and seasonal fluxes), a two-way analysis of variance for normally distributed data and the Friedman test for non-normally distributed data was used. However, no interaction between forest type and drying-rewetting cycles was found.

## 3. Results

*3.1. Soil Water Dynamics and Soil Temperature*

During the monitoring period, the mean soil temperature at a depth of 15 cm varied between 0.3 and 20.4 °C (Figure 1a), while air temperatures ranged from −9.9 to 31.3 °C. The soil temperature was not significantly influenced by the admixture of fir into beech forests. Soil WFPS also hardly differed between BB and BF except for a dry period in summer 2016, during which higher WFPS values were observed at BF compared to BB. The monitored years differed in rainfall regimes. In 2016, there was a very wet spring and early summer, followed by a drought period in August/September with strongly decreasing WFPS in the measured depth of 15 cm. In contrast, 2017 showed a more uniform pattern of precipitation distribution over the growing season, resulting in less pronounced temporal changes of soil moisture throughout the growing season. Generally, high WFPS values were observed in the dormant season.

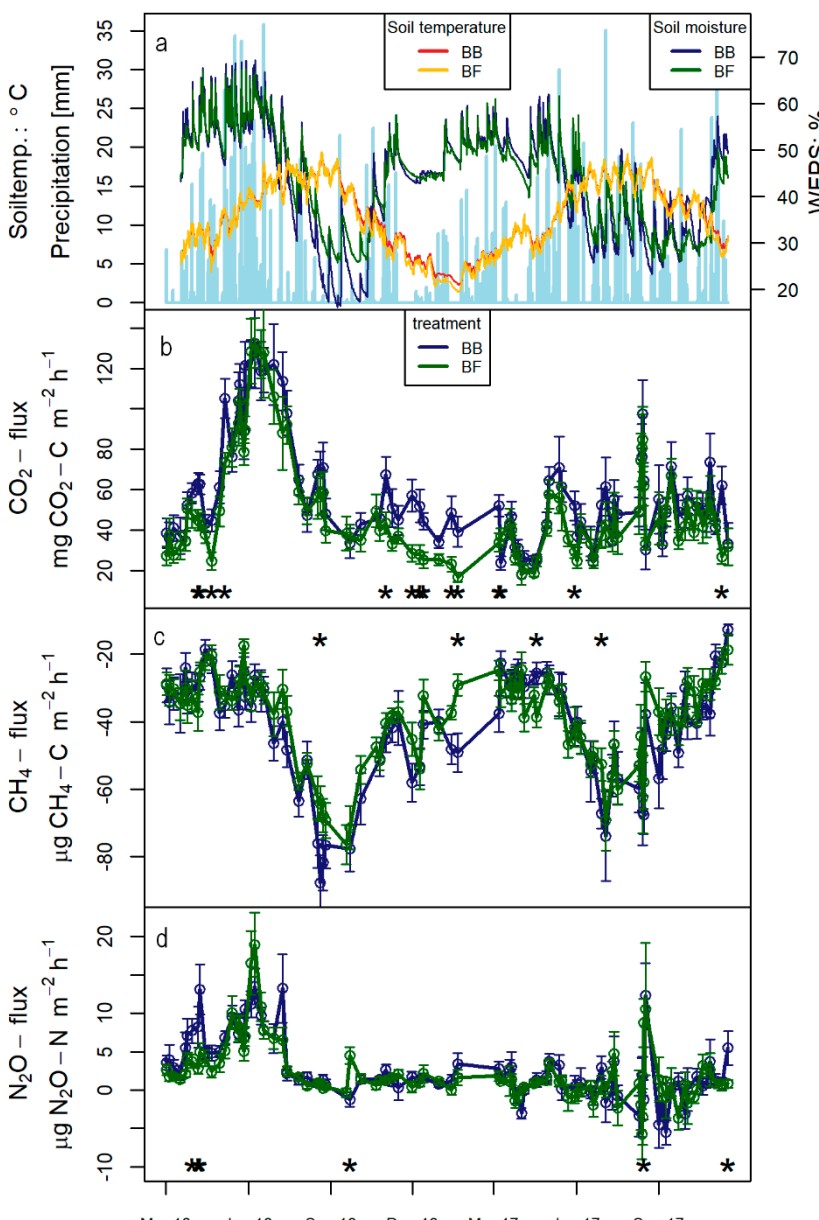

**Figure 1.** Soil temperature and soil moisture (WFPS: Water filled pore space) (**a**) and the soil-atmosphere exchange of $CO_2$ (**b**), $CH_4$ (**c**), and $N_2O$ (**d**) in the pure beech (BB) and mixed beech fir stands (BF) in the period March 2016 to November 2017: The graph shows the treatment means with standard errors ($n$ = 5–6). The asterisks indicate significant differences between BB and BF at given sampling dates (Wilcoxon–Mann–Whitney test, $p < 0.05$).

Rainout shelters reduced soil moisture in the topsoil during both summers to an average minimum (BB drought and BF drought) of 17.5% water-filled pore space (9.6 $m^3m^{-3}$ volumetric water content) (Figure 2a). Following the rewetting event at the end of the simulated drought period in 2017, the soil water-filled pore space increased from 24 to 35% and from 21 to 43% under BB and BF, respectively.

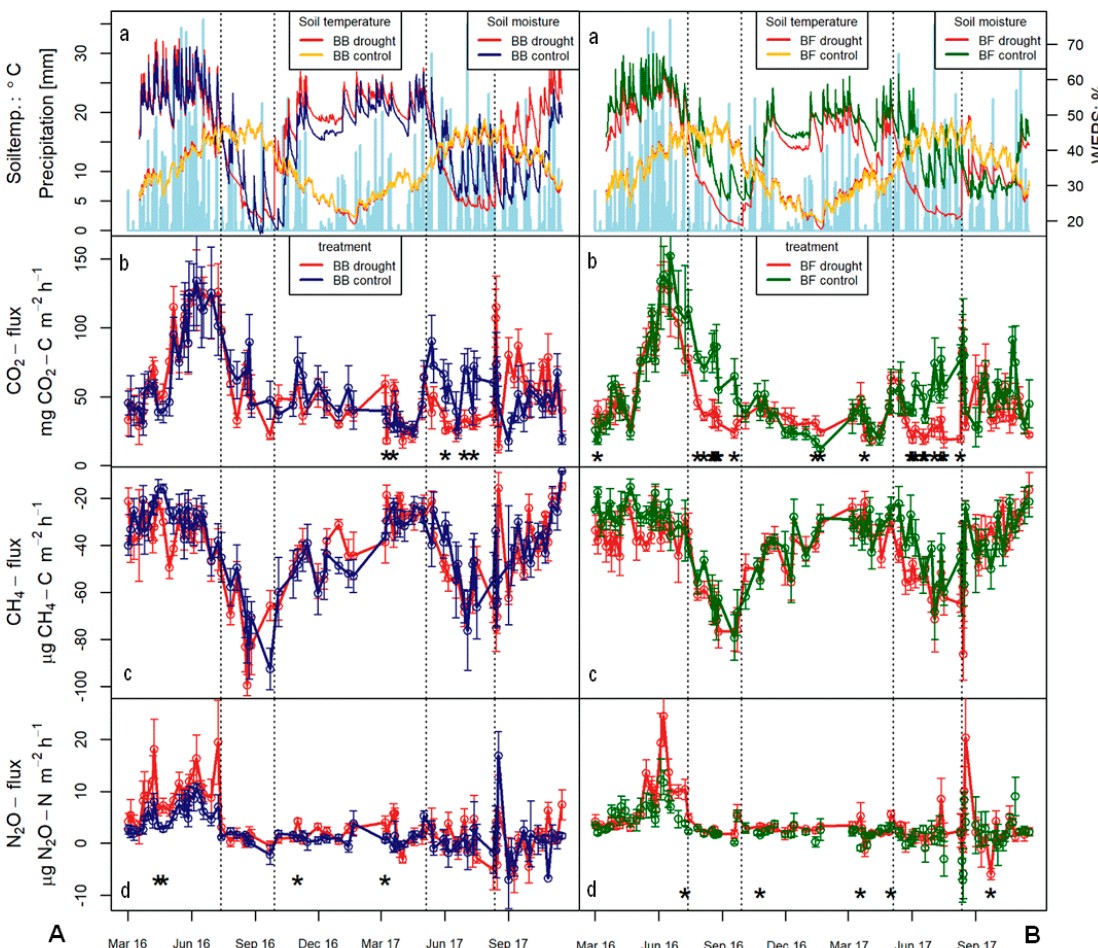

**Figure 2.** Soil temperature and soil moisture (**a**) and the soil-atmosphere exchange of $CO_2$ (**b**), $CH_4$ (**c**) and $N_2O$ (**d**) as influenced by rainout shelters in summer for pure beech stands (**A**) and mixed beech-fir stands (**B**): The dotted lines indicate the start and end of the simulated drought period in summer (BF/BB drought). The uncertainty is given as the standard error of the mean (*n* = 5–6). WFPS: Water filled pore space. The asterisks indicate significant differences between the BB/BF control and the BB/BF drought at the given sampling dates (Wilcoxon–Mann–Whitney test, *p* < 0.05).

*3.2. Effects of the Admixture of Fir into Beech Forests on the Soil-Atmosphere Exchange of GHGs*

The mean soil respiration rates as measured for the treatments of this study ranged between 10 and 150 mg $CO_2$–C m$^{-2}$ h$^{-1}$ with the highest rates being observed during the wet early summer of 2016 (Figure 1). The highest seasonal cumulative rates were reached in the growing season (1st May–31st October) 2016 with 3.3 ($\pm$0.3) and 3.0 ($\pm$0.3) t C ha$^{-1}$ for BB and BF, respectively, with significantly smaller soil respiration rates in the following growing season and the dormant season (Figure 3). The admixture of fir significantly decreased soil respiration in autumn and winter (Figure 1), which resulted in a substantial difference in the cumulative soil respiration rates between BB (1.8 $\pm$ 0.1 t C ha$^{-1}$) and BF (1.2 $\pm$ 0.0 t C ha$^{-1}$) during the dormant season (1st November–30th April, Figure 3). During the growing seasons, we observed no influence of the fir admixture. However, no significant influence (*p* = 0.08) of the fir admixture on soil respiration was found for the annual cumulative $CO_2$ flux rates (29th March, 2016 to 28th March, 2017) (Table 1).

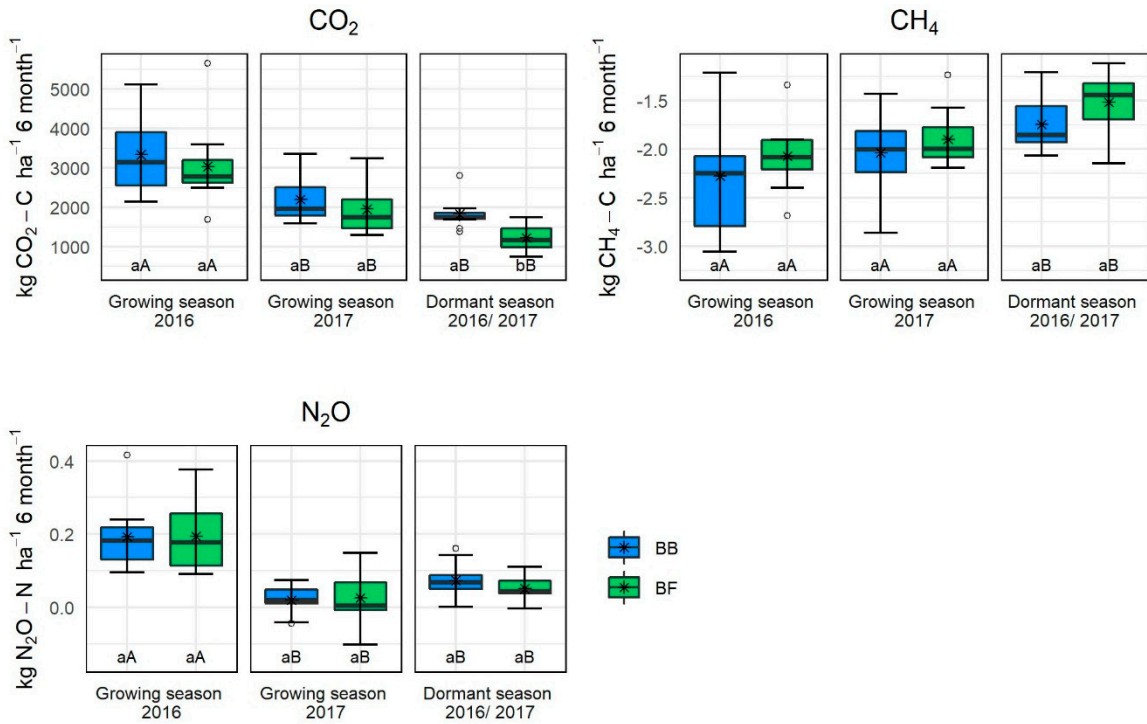

**Figure 3.** The cumulative fluxes of $CO_2$, $CH_4$ and $N_2O$ for the growing seasons of 2016 and 2017 and the dormant season 2016/2017 for pure beech (BB, blue) and mixed beech-fir (BF, green) stands: The values are shown as boxplots with medians, the 25th and 75th percentile, the most extreme data points (whiskers´ extension), the outliners as unfilled circles as well as the means as asterisks (*n* = 5–6). The small letters indicate significant differences between BB and BF; capital letters indicate significant differences between the seasons (normally distributed data: t-test/ANOVA and Tukey post hoc test; non-normal distributed data: Wilcoxon–Mann–Whitney test/Kruskal–Wallis and Bonferroni post hoc test, *p* < 0.05).

The average net $CH_4$ uptake rates ranged between 8 and 99 µg $CH_4$–C $m^{-2}$ $h^{-1}$ and showed a clear seasonal pattern with the greatest uptake rates during the drier and warmer summer months and the smallest uptake rates in the dormant season (November–April) (Figures 1 and 3). At some single measurement points, the pure beech stand revealed significantly higher net $CH_4$ uptake rates than the mixed stand. However, this did not translate into significant differences in the annual cumulative $CH_4$ fluxes between BB ($-4.0 \pm 0.3$ kg C $ha^{-1}$ $year^{-1}$) and BF stands ($-3.6 \pm 0.2$ kg C $ha^{-1}$ $year^{-1}$) (Table 1). Also, the cumulative seasonal $CH_4$ fluxes were not influenced by fir admixture (Figure 3).

With exchange rates between $-9.4$ and 25.3 µg $N_2O$–N $m^{-2}$ $h^{-1}$, $N_2O$ fluxes were very small (Figure 1) and hardly showed seasonal patterns. The highest $N_2O$ emissions were observed—similar to soil respiration rates—in the wet early summer of 2016 (Figure 1). Hence, the growing season 2016 showed with 0.19 ($\pm$ 0.03) kg N $ha^{-1}$ (both for BB and BF stands), significantly higher cumulative $N_2O$ emission rates than the dormant season (0.07 ($\pm$ 0.02) (BB), 0.05 ($\pm$ 0.01) kg N $ha^{-1}$ (BF)). Fluxes during the growing season 2017 overall were not significantly different from zero both for BB and BF stands (Figure 3).

**Table 1.** The annual cumulative flux rates (March 2016–March 2017) of soil respiration ($CO_2$), $CH_4$ and $N_2O$; the related global warming potentials (GWP) of $CH_4$ and $N_2O$; and the non-$CO_2$ greenhouse gas (GHG) balance of soils of the pure beech (BB) and mixed beech-fir stand (BF) stands; mean ($\pm$ se). Furthermore, the soil organic carbon (SOC) stocks for the organic horizon and the top mineral soil (0–7.5 cm) as well as for the entire soil profile are presented ($n = 3$, mean $\pm$ se). Indices indicate significant differences within the horizon ($p < 0.05$). The annual SOC stock changes (SOC stocks GWP) due to fir admixture are calculated for a stand age of 50 years. These data are merged with changes in non-$CO_2$ GWP due to fir admixture to calculate the total soil GHG budget.

| | Soil respiration (kg C $ha^{-1}year^{-1}$) | $CH_4$ (kg C $ha^{-1}year^{-1}$) | $N_2O$ (kg N $ha^{-1}year^{-1}$) | $CH_4$ GWP (kg $CO_2$-eq $ha^{-1}year^{-1}$) | $N_2O$ GWP (kg $CO_2$-eq $ha^{-1}year^{-1}$) | Non-$CO_2$ GHG balance (kg $CO_2$-eq $ha^{-1}year^{-1}$) |
|---|---|---|---|---|---|---|
| BB | 5347 ($\pm$323) | −4.0 ($\pm$0.3) | 0.32 ($\pm$0.05) | −150 ($\pm$10) | 132 ($\pm$20) | −18 |
| BF | 4398 ($\pm$372) | −3.6 ($\pm$0.2) | 0.27 ($\pm$0.04) | −133 ($\pm$7) | 112 ($\pm$15) | −21 |
| | SOC stock organic horizon (t $ha^{-1}$) | SOC stock 0–7.5 cm (t $ha^{-1}$) | SOC stocks organic + 7.5 cm profile (t $ha^{-1}$) | SOC stocks organic + 90 cm profile (t $ha^{-1}$) | GWP of increased SOC storage at BF (0–90 cm) (kg $CO_2$-eq $ha^{-1}year^{-1}$) | Total soil GHG budget of fir admixture (kg $CO_2$-eq $ha^{-1}year^{-1}$) |
| BB | 15.3 ($\pm$0.4) a | 22.2 ($\pm$2.5) a | 36.8 ($\pm$2.0) | 87.9 ($\pm$5.1) | | |
| BF | 20.6 ($\pm$0.7) b | 29.2 ($\pm$3.2) b | 49.5 ($\pm$4.0) | 97.0 ($\pm$5.5) | | |
| Difference BF-BB | | | 12.8 | 9.1 | −664 | −667 |

*3.3. Effects of Drying-Wetting Cycles on the Soil-Atmosphere Exchange of GHGs and Interactions with Fir Admixture*

The simulated summer drought significantly decreased soil respiration under both forest types (Figure 2). During the drought period 2016, this was more pronounced under BF where we found higher differences in mean soil respiration rates between the drought (38 mg $CO_2$–C $m^{-2}$ $h^{-1}$) and control treatments (74 mg $CO_2$–C $m^{-2}$ $h^{-1}$) compared to BB (57 and 68 mg $CO_2$–C $m^{-2}$ $h^{-1}$ for drought and control, respectively). In 2017, the roof had a similar effect on the mean soil respiration in the beech and mixed beech-fir stand (BB: 60 and 33 mg $CO_2$–C $m^{-2}$ $h^{-1}$ for control and drought; BF: 53 and 27 mg $CO_2$–C $m^{-2}$ $h^{-1}$ for control and drought, respectively). An intensified drought significantly reduced cumulative soil respiration in the simulated drought periods in the mixed stand (both years) and pure beech stand (2016) (Table 2). The rewetting event at the end of the simulated drought period in 2017 was followed by a sudden increase in $CO_2$ fluxes from 38 mg to 115 mg $CO_2$–C $m^{-2}$ $h^{-1}$ under BB and from 18 mg to 82 mg $CO_2$–C $m^{-2}$ $h^{-1}$ under BF, which were the highest fluxes in 2017. For both forest stands, the maximum in soil respiration was reached one day after rewetting. Fluxes dropped to pre-wetting levels at around the fourth day after rewetting.

In both forest stands, neither the uptake of $CH_4$ nor the release of $N_2O$ differed significantly between the drought and control plots during the simulated drought periods (Figure 2, Table 2). Generally, rewetting induced a decrease in net $CH_4$ uptake (Figure 2). E.g., the rewetting event in 2017 caused a decline in net-$CH_4$ uptake rates from approx. 70 to 20 mg $CH_4$–C $m^{-2}$ $h^{-1}$ within two days both in mixed and pure beech stands. Nitrous oxide fluxes in both stands showed only minor and short-lived responses to rewetting in 2017 and no response in 2016, so that the drying-wetting cycles did not translate into altered seasonal fluxes (Table A2).

**Table 2.** The cumulative flux rates of the simulated drought periods (2016: 78 days; 2017: 99 days) of soil respiration ($CO_2$), $CH_4$ and $N_2O$ of the control as well as drought treatment of the pure beech (BB) and mixed beech-fir stand (BF); mean ($\pm$se). Significant differences are indicated by different letters at $p < 0.05$.

| | BB Control | BB Drought | *p* Value | BF Control | BF Drought | *p* Value |
|---|---|---|---|---|---|---|
| | $CO_2$ (kg C $ha^{-1}$ $season^{-1}$) | | | | | |
| Drought period 2016 | 1169 ($\pm$97) | 1004 ($\pm$33) | 0.548 | 1341 ($\pm$62) a | 718 ($\pm$26) b | 0.025 |
| Drought period 2017 | 1465 ($\pm$97) a | 853 ($\pm$28) b | 0.016 | 1276 ($\pm$62) a | 664 ($\pm$44) b | 0.030 |
| | $CH_4$ (kg C $ha^{-1}$ $season^{-1}$) | | | | | |
| Drought period 2016 | −1.7 ($\pm$0.1) | −1.6 ($\pm$0.1) | 0.829 | −1.5 ($\pm$0.1) | −1.5 ($\pm$0.4) | 0.815 |
| Drought period 2017 | −1.2 ($\pm$0.1) | −1.2 ($\pm$0.0) | 0.846 | −1.0 ($\pm$0.0) | −1.3 ($\pm$0.0) | 0.082 |

**Table 2.** *Cont.*

|  | BB Control | BB Drought | *p* Value | BF Control | BF Drought | *p* Value |
|---|---|---|---|---|---|---|
| $N_2O$ (kg C ha$^{-1}$ season$^{-1}$) | | | | | | |
| Drought period 2016 | 0.15 (±0.02) | 0.27 (±0.06) | 0.503 | 0.15 (±0.02) | 0.30 (±0.02) | 0.846 |
| Drought period 2017 | 0.06 (±0.04) | 0.10 (±0.06) | 0.138 | −0.12 (±0.05) | 0.23 (±0.03) | 0.093 |

### 3.4. Environmental Controls of Soil-Atmosphere GHG Fluxes

Both soil temperature and moisture showed positive relationships with the soil respiration of BB and BF stands (Figure 4), yet with low explanatory power of 11%–20% only. Multiple regression revealed that both indicators were significant and together explained 58% (BB) to 62% (BF) of the variance of soil respiration. Soil moisture was the dominant control of $CH_4$ fluxes with an increasing net $CH_4$ uptake at decreasing soil moisture ($R^2$ = 0.42 to 0.50) (Figure 4). The net $CH_4$ uptake also increased with increasing temperatures. For the BB stand, soil temperature was excluded by the multiple regression approach (Table 3). Soil $N_2O$ fluxes were not related to soil temperature for BB and only a little for BF ($R^2$ = 0.04, $p < 0.05$) but increased at soil moisture values > 25 vol %.

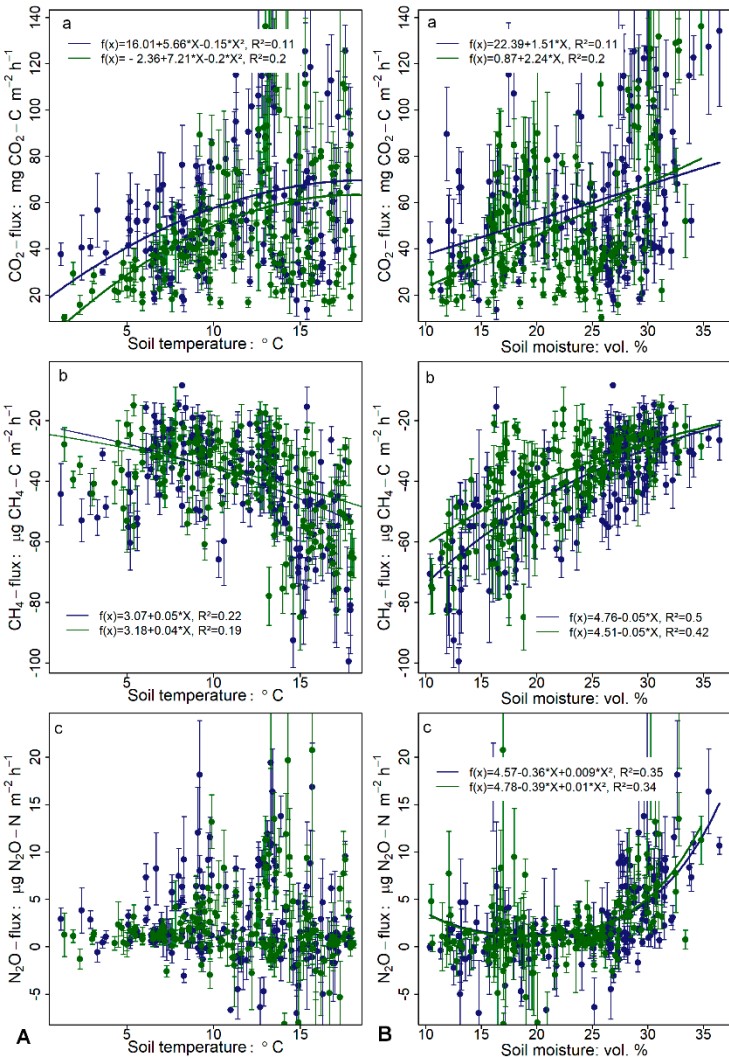

**Figure 4.** The regressions of soil respiration (**a**), $CH_4$ fluxes (**b**) and $N_2O$ fluxes (**c**) with soil temperature (**A**) and soil moisture (**B**): Presented are the means of the four different plots (BB control and drought: blue, BF control and drought: green, *n* = 5–6) on a sampling day with standard errors. The data show 91 measurements from March 2016 to November 2017.

**Table 3.** The regression analyses to disentangle the relationships between soil temperature/moisture and soil-atmosphere GHG fluxes: The table shows both regressions for the two single factors (as illustrated in Figure 4) as well as multiple regression models to illustrate the combined explanatory power of moisture and temperature ($n = 91$ each). The significant effects are shown in bold at $p < 0.05$.

| | $CO_2$ | | | | $CH_4$ | | | | $N_2O$ | | | |
| | BB | | BF | | BB | | BF | | BB | | BF | |
| | $R^2$ | $p$ | $R^2$ | $p$ | $R^2$ | $p$ | $R^2$ | $p$ | $R^2$ | $p$ | $R^2$ | $p$ |
|---|---|---|---|---|---|---|---|---|---|---|---|---|
| Soil moist. | **0.11** | **<0.0001** | **0.20** | **<0.0001** | **0.50** | **<0.0001** | **0.42** | **<0.0001** | **0.35** | **<0.0001** | **0.34** | **<0.0001** |
| Soil temp. | **0.10** | **<0.0001** | **0.20** | **<0.0001** | **0.22** | **<0.0001** | **0.19** | **<0.0001** | 0.01 | 0.134 | **0.04** | **0.015** |
| Overall model | **0.58** | **<0.0001** | **0.62** | **<0.0001** | **0.50** | **<0.0001** | **0.47** | **<0.0001** | **0.43** | **<0.0001** | **0.36** | **<0.0001** |

*3.5. SOC Stocks*

We found significantly larger SOC stocks in the organic layer under BF with 20.6 ($\pm$ 0.7) t SOC ha$^{-1}$ compared to BB (15.3 $\pm$ 0.4 t SOC ha$^{-1}$) (Table 1). Also, the uppermost 7.5 cm of mineral soil revealed significantly higher SOC stocks at BF (29.2 $\pm$ 3.2 t SOC ha$^{-1}$) compared to BB (22.2 $\pm$ 2.5 t SOC ha$^{-1}$). Together, this translates into SOC stock increases of 12.8 t ha$^{-1}$ due to fir admixture. In contrast, there were no significant differences in SOC stocks between BB and BF stand in deeper horizons (7.5–15 cm, 15–30 cm, 30–50 cm and 50–90 cm). Considering the entire soil profile of 90 cm, this resulted in a similar SOC stock increase of 9.1 t C at BF (Table 1).

*3.6. Total Soil GHG Budget*

The negative global warming potential, expressed as $CO_2$ equivalents, of the biological net $CH_4$ sink slightly exceeded the positive global warming potential of the net $N_2O$ emissions so that the non-$CO_2$ GHG budget was slightly negative but with $-18$ and $-21$ kg $CO_2$-eq ha$^{-1}$ year$^{-1}$ for BB and BF, respectively, hardly different from zero (Table 1). Hence, fir admixture did not significantly affect the non-$CO_2$ GHG balance of the soil. Based on the difference in SOC stocks between BF and BB, we calculated an additional SOC sequestration of 9.1 t ha$^{-1}$ due to fir admixture during the time of stand development of 50 years. This equaled an improved global warming potential of $-664$ kg $CO_2$-eq ha$^{-1}$year$^{-1}$ of BF compared to BB. Considering the minimal decrease in the non-$CO_2$ global warming potential of $-3$, the total GWP of fir admixture is $-667$ kg $CO_2$-eq ha$^{-1}$ year$^{-1}$, i.e., an additional soil GHG sink of more than half a ton of $CO_2$ per ha and year.

## 4. Discussion

*4.1. Effects of Fir Admixture into Pure Beech Forests on Soil-Atmosphere GHG Exchange and SOC Stocks*

### 4.1.1. Soil Respiration and SOC Stocks

The soil respiration revealed significantly lower rates for the mixed beech-fir stand than the pure beech stand during the dormant season (Figures 1 and 3), thereby providing an explanation of the increased SOC sequestration due to fir admixture via the decreased decomposition of soil organic matter. The annual soil respiration measurements of 4.4–5.4 t C ha$^{-1}$ resemble those from previous measurements in temperate European beech forests, where rates between 2.0 and 8.7 t C ha$^{-1}$year$^{-1}$ [23,27,31,47] have been reported. Research on soil respiration under silver fir and mixed coniferous/deciduous stands are rare. In Douglas fir stands, soil respiration rates between 1.2 (Mediterranean forest, Spain) [27] and 13.7 t C ha$^{-1}$year$^{-1}$ (temperate forest on volcanic soil, Washington, USA) [46,48] were measured. Borken and Beese [31] found in mixtures of spruce and beech intermediate rates of the pure stands. A study conducted in central Italy [28] examined the soil respiration rates of pure silver fir and pure European beech by monthly measurements. The calculated annual fluxes reached values of 9.97 under beech and 12.41 t C ha$^{-1}$ year$^{-1}$ under pure fir, thus surprisingly revealing lower rates under deciduous tree species. In contrast, other

studies generally found that deciduous forests have higher soil respiration rates than coniferous stands [31], on average 10% [24], which is in line with our study. This might be due to higher summer soil temperatures of up to 2 °C in the fir stands investigated by Certini et al. [28], whereas the soil temperatures in our study were equal for pure beech and mixed beech-fir. From September to April, monthly flux rates were similar under these two Mediterranean forest types, [28] whereas pure beech soils at our study site revealed significantly higher soil respiration rates than the mixed beech-fir stand during autumn and winter (Figure 1). This is most likely a consequence of changes in litter quantity and quality after the input of organic matter via leaf fall. In beech forests, $CO_2$ fluxes reduced by 30% after litter removal was found [49], suggesting a high influence of the forest floor on soil respiration. Raich and Tufekciogul [24] also point to better decomposable leaf litter and faster nutrient cycling rates under deciduous species. We conclude that—at least for the temperate forests—the admixture of fir into pure beech stands leads to reduced soil respiration, thereby reducing carbon losses. This is in agreement with measurements of SOC stocks at our study site: We found a significant enrichment in carbon in the mixed beech-fir stands in the organic horizon and the top mineral soil (0–7.5 cm) (Table 1). This is in line with earlier work which showed a significantly positive effect of the admixture of coniferous trees (Douglas fir, spruce) on SOC stocks in the forest floor and mineral topsoil of temperate beech forests [50]. While the very high forest floor SOC stocks in coniferous forests might be vulnerable to disturbance or temperature increases, the SOC stocks in both forest floor and mineral soil of mixed deciduous/coniferous stands appear to be more resilient to disturbance and climate change [51]. SOC stocks are influenced by C input through root and leaf litter and rhizodeposition [24,49,52]. It has been shown that the age of carbon residues in recently fallen litter as well as in the soil of deciduous forests is lower than in coniferous forests [53], indicating a faster turnover. Higher base saturation, calcium concentrations and pH values are generally found under broadleaf tree species [50,54], thereby facilitating higher microbial decomposition and associated C output. In our study, soil respiration at BF was reduced by approx. 0.95 t C $ha^{-1}$ $year^{-1}$ compared to BB, while the annual net SOC stock increase amounted to approx. 0.2 t C $ha^{-1}$ $year^{-1}$ over 50 years. Considering that roughly half of soil respiration is heterotrophic respiration [55], the decreased $CO_2$–C loss from microbial respiration is larger than the long-term mean gain in SOC. Even if comparability is limited due to the strongly different timescales, this could indicate that under fir admixture, not only C losses but also C inputs are reduced, thereby still resulting in a net SOC increase.

### 4.1.2. $CH_4$ Fluxes

The investigated soils acted as net $CH_4$ sinks during the entire study period (Figure 1), as typical for upland forest soils [20,26]. The soil $CH_4$ uptake rates measured in this study are in the mean range of fluxes measured in northern European temperate forests [26,56,57]. While in our study, the net $CH_4$ sink strength was only insignificantly higher in BB than BF stands, generally deciduous forests revealed higher $CH_4$ oxidation rates than coniferous stands [32,58]. The biological $CH_4$ sink in deciduous forests can be even twice as high compared to coniferous forest [20,25–27]. Borken and Beese [20] investigated pure stands of beech and spruce as well as their mixtures with 30% spruce or beech, respectively. They detected the largest differences between the pure spruce and all other stands (pure beech and the two mixtures). Pure beech and the two mixtures were not consistently different, indicating a rather small influence of spruce [20]. This was similar in our study with only slightly lower $CH_4$ uptake rates due to fir admixture. Negative correlations between $CH_4$ oxidation and litter mass indicate that the forest floor acts as an additional diffusion barrier for $CH_4$ [56]. Indeed, in our study, the litter layer also in the beech-fir mixed stands still was dominated by beech leaves. However, the litter mass significantly differed with higher values under BF (4.6 kg $m^{-2}$) compared to BB (3.5 kg $m^{-2}$) ($p = 0.0001$). Methane availability is largest in the O horizon, but maximum consumption is usually in the top mineral soil [20]. Nevertheless, it was shown that beech Oa horizons have the capability to oxidize $CH_4$ and even maximum oxidation rates were found here, while for spruce, it was in the upper mineral soil [25]. However, Borken et al. [26] and Degelmann et al. [25] concluded that the

differences in the $CH_4$ uptake rates under deciduous and coniferous soils did not result from alterations in structure and gas permeability of the organic layer. Rather, monoterpenes which are present in conifer roots and needles have the potential to reduce $CH_4$ consumption by a factor of three [25,59]. Overall it might be a mixture of all these influences which result in changes in the diversity and population of methanotrophs that lead to a reduction of the $CH_4$ sink strength due to admixture of coniferous trees [20]. In our study, the effect of fir admixture on the net $CH_4$ sink strength remained small likely because of insufficient changes in the composition and diffusion properties of the litter layer. With increasing fir admixture, the $CH_4$ sink strength could potentially decrease. However, this is not expected to result in significant changes in the overall global warming budget.

### 4.1.3. $N_2O$ Fluxes

Soils under both forest types frequently switched between temporal net $N_2O$ sources and sinks (Figure 1) with overall small emissions. The impact of this emission on the atmospheric radiative forcing was approximately balanced by the biological $CH_4$ sink in the soil (Table 1). In beech forests, $N_2O$ emissions between 0.04 [56] and 6.6 kg N ha$^{-1}$ year$^{-1}$ [33] have been reported so the emission in this study (BB: $0.32 \pm 0.05$ and BF: $0.27 \pm 0.04$ kg N ha$^{-1}$ year$^{-1}$) clearly is at the lower end of this range, probably due to the sandy soil texture with a small proportion of anaerobic microsites favoring denitrification [60]. A Douglas fir stand revealed $N_2O$ rates between 0.22 and 0.31 kg N ha$^{-1}$ year$^{-1}$ [27]. The relatively low rates at our site can be explained by nutrient limitation and high carbon–nitrogen ratios [22]. No clear effect of the admixture of fir was observed in our study (Figure 1) which is in line with Borken and Beese [20], observing no clear effect of spruce admixture to beech on $N_2O$ emissions. However, between pure beech and spruce stands, Butterbach-Bahl et al. [33] found a great difference in the predominant trace gas emitted at high N availability: For beech, the mean annual emission rates of $N_2O$ were up to four times higher compared to spruce. For spruce, nitric oxide was the dominant N-trace gas. A main factor influencing $N_2O$ fluxes in soils is the litter layer. After litter removal, $N_2O$ fluxes 118% lower in soils were measured [49]. Especially after rainfall, the litter layer acts as a diffusion barrier for oxygen and, therefore, creates anaerobic microsites which favor $N_2O$ production [20,49]. Especially wet beech leaves tend to adhere, thereby creating anoxic conditions in the organic layer [20] that explains higher emissions under deciduous forest stands. In mixtures of beech with fir, the still dominating beech leaves in the forest floor seem to be the main explanation for missing differences in $N_2O$ emissions due to fir admixture.

### 4.2. Drying-Wetting Cycles—Effects on Soil Respiration and $CH_4$ and $N_2O$ Fluxes

### 4.2.1. Soil Respiration

Soil respiration was limited not only by low temperatures but also by low moisture (Figure 4a). Hence, simulated summer droughts significantly reduced soil respiration both at BB (2017) and BF (2016 and 2017) (Figure 2). Our data on soil water dynamics suggest that the water availability in the soil of pure beech stands is more prone to drought than that of the mixed stand: Soil moisture in the pure beech stand declined much faster and during less intensive droughts and was more intensively affected by the natural drought in 2016 (Figure 2A: BB control). The reason might be the anisohydric habits of beech that maintain high transpiration rates, also during drought, [17,61] whereas firs react more sensible with the early closure of the stomata [62]. Therefore, admixing fir to pure beech could maintain a higher soil moisture during short-term drought periods.

The effects of drought on soil respiration and its heterotrophic microbial contribution are discussed throughout the literature, and often a strong reduction in $CO_2$ flux from soils is reported [32,35–37]. This can be explained by the reduction of the biological activity of microorganisms through drought stress, either by dehydration (i.e., dormant state) or by the acclimation of microorganisms at high costs [63,64] and even a dieback of microorganisms [65]. Further, a limited availability of carbon due to less substrate diffusion during reduced soil water content limits the accessibility of substrates for

microorganisms [37]. The rewetting event in 2017 led to a sharp increase of $CO_2$ emissions under both forest types with an increase of up to four times the pre-wetting emissions and even reaching maximum rates in 2017. This was in line with Xu and Luo [37] and with Borken et al. [26], detecting increased soil respiration with higher wetting intensity. The sudden rewetting pulse of soil respiration was most likely due to enhanced microbial activity shifting from a dormant to active state [66] as well as the rapid regrowth and changes in the composition of microbial communities [37]. As the quick increase in soil respiration shows, microorganisms are well adapted to regular drying-wetting cycles [35]. However, physical effects like the replacement of $CO_2$ from soil pores through rewetting cannot be excluded and might also have contributed to high $CO_2$ peaks [56]. Nevertheless, recovered microbial activity seems to play a great factor, since Xu and Luo [37] and Borken et al. [26] reported a rather quick decrease in soil labile organic C after some days showing a flush of C mineralization. Heavy rainfalls on dry soil, therefore, can counterbalance the reducing effect of summer droughts on soil respiration. Assuming that microbial respiration accounted for half of soil respiration [55], in our study, a great reduction of C mineralization during drought was found which was not completely counterbalanced by rewetting (Table A2), pointing to a reduced C loss in theses forest soils under climate change conditions. All of this seemed little affected by fir admixture so that the resilience of soil respiration in beech stands to drying-wetting cycles in the growing season overall appears little affected by fir admixture.

### 4.2.2. $CH_4$ and $N_2O$ Fluxes

During the simulated drought period, $CH_4$ and $N_2O$ fluxes did not differ significantly between the drought and control plots under both stands (Figure 2, Table 2). This was rather unexpected since both GHGs are known to be regulated by soil moisture [22]. However, in 2016, moisture regimes were rather similar for the drought and control plots because the relatively wet spring and early summer were followed by very dry weather, which strongly decreased soil moisture in the mineral soil of control plots without rainout shelter as well. Hence, the $CH_4$ uptake increased both in the soil of control and drought plots due to an improved diffusion of $CH_4$ and $O_2$ into the soil. The rewetting (2017) of the dry soil initiated an increase in $CH_4$ uptake on the first couple of days which was followed by a sharp decline after 4–5 days. This was similar for the BB and BF stand. Other studies reveal contradicting results showing an immediate decline in $CH_4$ uptake with the intensity of wetting without any prior increase in $CH_4$ oxidation [37]. This can be explained by the high water table, which blocks diffusion within the soil matrix, resulting in less oxidation [22,37]. In summary, we conclude that drought generally increases the $CH_4$ sink strength; however, this is limited by drought stress for microorganisms at extremely low soil water availability in the topsoil.

Nitrous oxide fluxes—being generally low—were equally small during the simulated drought periods in all stands. Especially in 2016, drought obviously diminished the activity of nitrifying and denitrifying microorganisms in line with previous work of Muhr et al. [36]. Rewetting either caused no effect or an increase of the flux rate (BF, one chamber). Several authors reported a sharp increase in $N_2O$ fluxes after wetting [37,40], explained by a higher substrate availability, an enhanced microbial metabolism and anaerobic conditions [34,40]. However, others, e.g., Reference [36], reported no significant effect after rewetting with fluxes remaining close to zero as observed in most cases of our study. The soil´s texture is assumed to be responsible for these differences: Our soils and those investigated by Muhr et al. [36] are sandy loams where a faster drainage is expected and especially after a long drought period, conditions within the soil matrix most likely stay aerobic. After rewetting in August 2017, the soil reached an average WFPS of only 35% (BB) and 43% (BF). Overall, $CH_4$ and $N_2O$ fluxes were generally very low at our study site and even intense drying-wetting cycles only marginally changed $N_2O$ fluxes both in pure beech and mixed beech-fir stands.

## 5. Perspectives for Forest Management and Outlook

Our case study based on almost two years of monitoring of the soil-atmosphere exchange of greenhouse gases showed that for the total soil GHG balance, nitrous oxide and methane played a negligible role while changes in C sequestration in the soil were dominant. The admixture of fir to European beech forests significantly reduced soil respiration likely due to the more recalcitrant fir litter, thereby contributing to larger SOC stocks in beech-fir mixtures. Consequently, fir admixture will increase the soil GHG sink strength of these forests, while no changes regarding drying-wetting events are to be expected. The data of our study, therefore, suggests that also under future climate conditions, soils of mixed beech-fir stands will represent a larger GHG sink than those of pure beech stands through an increased SOC stock. These findings are relevant for predicting GHG fluxes under different silvicultural practices as well as estimating the carbon balance of forest soils under the auspices of climate change. Furthermore, since increased SOC stocks generally improve key soil functions and ecosystem services such as water and nutrient retention, filter capacity, erosion control and productivity, fir admixture is recommended to increase the sustainability of beech forests in a changing climate.

**Author Contributions:** Conceptualization: M.D. and H.R.; performance of experiments: S.R., M.F., J.T., R.-K.M., T.B. and M.D.; laboratory work: S.R., M.F., J.T. and A.S.-S.; data curation: S.R., M.F., J.T., A.S.-S. and M.D.; investigation: S.R., M.F., J.T. and R.-K.M.; project administration: T.B., M.D. and H.R.; funding acquisition: M.D. and H.R.; supervision: M.D. and H.R.; visualization: S.R. and M.F.; writing—original draft: S.R. and M.F.; writing—review and editing: M.D. and H.R.

**Funding:** This study was financially supported via the Bundesanstalt für Landwirtschaft und Ernährung (BLE), Germany, by the Bundesministerium für Ernährung und Landwirtschaft (BMEL) and the Bundesministerium für Umwelt, Naturschutz, Bau und Reaktorsicherheit (BMUB) within the program "Waldklimafonds" (No. 28W-C-1-069-01) in the frame of the project "Buchen-Tannen-Mischwälder zur Anpassung von Wirtschaftswäldern an Extremereignisse des Klimawandels (BuTaKli)", based on the decision of the German Federal Parliament. This article has been funded through the Open Access Publishing Fund of Karlsruhe Institute of Technology.

**Acknowledgments:** Our thank goes to Lukas Neumann, Bárbara San Martín and Lisa Wienkenjohann for their great help with the fieldwork. Furthermore, we thank the students of the course "Biosphere-atmosphere exchange and soil processes" (Paul Bauer, Anna Gäßler, Lara-Sophie Heitbrink, Susanne Hermann, Eva Leypold, Saskia Oehler and Isabell Schlangen) for their invaluable help with the soil sampling and analysis.

**Conflicts of Interest:** The authors declare no conflict of interest.

## Appendix A

**Table A1.** The stand characteristics of the stand in the Black Forest, Freiamt.

| Elevation (Meters above Sea Level) | 400–460 |
|---|---|
| Mean annual air temperature (°C) (period) | 9.6 (1971–2003) |
| Annual precipitation (mm) (period) | 1020 (1971–2003) |
| Stand age (years) | 40–60 |
| Growth rate (diameter at breast height) of beech/silver-fir (meters/year) | 0.0062/0.0074 |
| Tree height (pure/mixed) (average) (meters) | 24 |
| Vegetation composition in a distance of 10 m around soil pits | |
| Plot I BB (hilltop) | 100% beech |
| Plot I BF (hilltop) | 45% beech, 34% silver-fir, 16% larch, 5% birch |
| Plot II BB (upper slope) | 86% beech, 7% oak, 7% birch |
| Plot II BF (upper slope) | 58% beech, 38% silver-fir, 4% birch |
| Plot III BB (middle slope) | 100% beech |
| Plot III BF (middle slope) | 75% beech, 20% silver-fir, 3% larch, 2% birch |

**Table A2.** The seasonal cumulative flux rates of soil respiration ($CO_2$), $CH_4$ and $N_2O$ of the control (no roof) as well as drought treatment (roof) of the pure beech (BB) and mixed beech-fir stand (BF); mean ($\pm$SE).

|  | **BB Control** | **BB Drought** | **BF Control** | **BF Drought** |
|---|---|---|---|---|
| **$CO_2$ (kg C ha$^{-1}$ season$^{-1}$)** | | | | |
| Growing S. 2016 | 3321 ($\pm$625) | 3347 ($\pm$243) | 3473 ($\pm$565) | 2675 ($\pm$250) |
| Growing S. 2017 | 2352 ($\pm$374) | 2054 ($\pm$162) | 2238 ($\pm$273) | 1716 ($\pm$280) |
| **$CH_4$ (kg C ha$^{-1}$ season$^{-1}$)** | | | | |
| Growing S. 2016 | –2.17 ($\pm$0.36) | –2.39 ($\pm$0.15) | –1.98 ($\pm$0.18) | –2.15 ($\pm$0.12) |
| Growing S. 2017 | –2.02 ($\pm$0.29) | –2.05 ($\pm$0.19) | –1.76 ($\pm$0.14) | –2.01 ($\pm$0.09) |
| **$N_2O$ (kg N ha$^{-1}$ season$^{-1}$)** | | | | |
| Growing S. 2016 | 0.15 ($\pm$0.02) | 0.24 ($\pm$0.05) | 0.14 ($\pm$0.03) | 0.24 ($\pm$0.04) |
| Growing S. 2017 | 0.03 ($\pm$0.01) | 0.01 ($\pm$0.02) | 0.00 ($\pm$0.03) | 0.05 ($\pm$0.03) |

The *t*-test (normal distributed data) and Wilcoxon test (non-normal distributed) revealed no significant differences at $p < 0.05$.

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
