# Peer review of "Admixing Fir to European Beech Forests Improves the Soil Greenhouse Gas Balance"

_forests, doi:10.3390/f10030213_

Round 1

Reviewer 1 Report

Authors present detailed soil GHG measurements from a beech stand and an adjacent intermixed fir-beech stand under natural conditions as well as under precipitation reduction/exclusion. Further, they performed some soil sampling at the two stands and compared soil C stocks. The topic of C sequestration and forest management is highly relevant and fits well within the scope of the journal. The paper is very well written. Authors did a very detailed literature review and discussed everything in great detail. On the one hand, this is fine; on the other hand it makes some passage a little lengthy, especially in the discussion. The discussion might be shortened a bit and only the parts which are really related to the findings in the study might be kept. Overall, there is room for further improvement, especially with regard to the soil C stock story.

Mayor comments:

With regard to GHG measurements, the study is important and convincing. GHG measurements were conducted in sufficient spatial and relatively high temporal resolution and the dataset is solid and very interesting, also with regard to precipitation manipulation. This really allows drawing some conclusions about the possible response of pure beech and fir intermixed stands to drought.

With regard to soil C stocks, the study is rather unconvincing. First problem is the lack in spatial replication of stands. As far as I understood, only two stands were compared. A (single) beech stand and an adjacent (single) fir-beech stand. Accordingly, the number of replicates is 1. The second problem is that there is no proof that the soil C stocks of the two stands were the same before switching one of them from beech to mixed fir-beech. It is clear that it is difficult because such old pre-treatment soil C data is hardly available, but with a replication of one stand it needs to be proven that C stocks were similar before treatment. If you had compared several stands in the region, the issue with pre-treatment stocks would have been of less concern. Overall, I suggest focusing more on the GHG story and only mention the soil C stocks as a simple site parameter and as an addition explanation parameter for the soil CO2 results (which indeed match quite well). Accordingly, the soil C stock story should be taken out of the title and the abstract.

In the title and in the whole manuscript it should be taken care to always making clear that the paper is about soil. Eg. In the Title it is written that “Admixing…improves the GHG balance…” For a reader it is not clear that the study is only about soil. A reader would think that NEP of the whole stand was assessed. This also happen a lot of times in the manuscript. Please be more specific e.g. by writing “Admixing…improves the soil GHG balance…” and so forth.

The site description is inadequate as no information about tree height, tree basal area, diameter, and most important, the mixing ratio of fir-beech is provided. You could add a little table and add the soil C stocks there already.

Specific comments:

Title, as suggested above, the soil C stock story is too weak for the title. The title may better be oriented towards the drought experiment and effects of tree-mixing during drought and effects on GHGs.

Abstract: The “research highlight” reads a bit strange. That you have measured GHGs and soil C stocks is not really a “research highlight”. However, I am also not sure what the journal wants to have written under this heading. “…under a changing climate” sounds strange a little, as well.

L23-24: Litter input and quality was not measured. Litter layer quantity is provided somewhere in the text. If you mean that, please write “litter layer” thickness or quantity. Quality as not assessed and therefore this speculation should not be made in the abstract.

L27: as mentioned above, this is highly misleading. The total GHG of forest conversion was never assessed! Please Write “the total soil GHG balance…”

There is no proof that mixing had any effects on soil C stocks – reasons above – so the soil C stock story should be removed from the abstract.

The introduction is fine. L51-54 This sentence may be re-phrased or split into two sentences. I did not understand the full meaning. L81 and elsewhere. You may add “Schindlbacher et al .GCB 2012 Soil respiration under climate change: prolonged summer drought offsets soil warming effects”. They also found slow recovery of soil CO2 after re-wetting in a quite similar mixed stand.

Methods: see above + L105: “…to fully exclude rainfall in order to induce summer drought” (it was not clear before the experiment that there would be natural drought already…

L108: “After two and a half months of rainfall exclusion in 2016, we applied 100 liters… and the same for the next sentence.

L112: Daily precipitation was measured at a nearby…

L118: company and country for the sensors as well

L123: if you had done this analyses at a couple of pure and mixed stands, it would have been ok for an analyses of management effects. The sampling presented here is just suitable for a site description.

L155: onwards. were the vials evacuated or filled with helium or just with air prior to sample injection?

L164: the long closure time and linear interpolation likely underestimates CO2 efflux

Results:

The chapters 3.2.1 and 3.3.1 (also3.2.2. and 3.3.2) should not have the exactly same headings. I’d suggest just putting them together into one. This would also ease reading – Or, if you decide keeping it separately, to change the headings so it is clear that the first one is about species effects in general and the second one about drought.

L212: better write the actual WFPS % number not relative %changes here.

Fig. 1 Is there any explanation for the relatively low CO2 efflux during summer 2017? Precipitation was regular and moisture not really low.

L274: ...which were the highest fluxes in 2017.

L275: ...dropped to pre-wetting levels at around the fourth day after rewetting.

Discussion:

L332: you can delete “measurements”

The whole chapter could be shortened a bit and be a little more focused. – as the whole discussion…

L438: ...during less intensive drought…was more intensively affected by…

L502: This is very un-specific. If you just write "resilience", a reader would think about resilience of the whole forest stand!

Author Response

Thank you very much for your critical and constructive feedback. Please find the responses to your remarks in the file attached.

Reviewer 2 Report

The manuscript is in good quality and the research topic is comprehensive. The author made a good development of the research methods and their findings are very importants to researchers in the area.

Author Response

Thank you very much for your feedback!

Round 2

Reviewer 1 Report

Fine paper. Very well revised.